DOI: 10.1038/s41467-017-01195-y · **OPEN**

# DNA N6-methyladenine is dynamically regulated in the mouse brain following environmental stress

Bing Yao[1], Ying Cheng[1], Zhiqin Wang[2], Yujing Li[1], Li Chen[1,5], Luoxiu Huang[1], Wenxin Zhang[3], Dahua Chen[3], Hao Wu[4], Beisha Tang[2] & Peng Jin [1]

Chemical modifications on DNA molecules, such as 5-methylcytosine and 5-hydroxymethylcytosine, play important roles in the mammalian brain. A novel DNA adenine modification, $N$(6)-methyladenine (6mA), has recently been found in mammalian cells. However, the presence and function(s) of 6mA in the mammalian brain remain unclear. Here we demonstrate 6mA dynamics in the mouse brain in response to environmental stress. We find that overall 6mA levels are significantly elevated upon stress. Genome-wide 6mA and transcriptome profiling reveal an inverse association between 6mA dynamic changes and a set of upregulated neuronal genes or downregulated LINE transposon expression. Genes bearing stress-induced 6mA changes significantly overlap with loci associated with neuropsychiatric disorders. These results suggest an epigenetic role for 6mA in the mammalian brain as well as its potential involvement in neuropsychiatric disorders.

[1] Department of Human Genetics, Emory University School of Medicine, Atlanta, GA 30322, USA. [2] Department of Neurology, Xiangya Hospital, Central South University, Changsha, Hunan 410008, China. [3] State Key Laboratory of Reproductive Biology, Institute of Zoology, Chinese Academy of Sciences, Beijing 100101, China. [4] Department of Biostatistics and Bioinformatics, Emory University School of Public Health, Atlanta, GA 30322, USA. [5] Present address: Department of Health Outcomes Research and Policy, Harrison School of Pharmacy, Auburn University, Auburn, AL 36849, USA. Bing Yao and Ying Cheng contributed equally to this work. Correspondence and requests for materials should be addressed to P.J. (email: peng.jin@emory.edu)

Covalent DNA modifications on the 5-carbon position of cytosine, such as 5-methylcytosine (5mC) and 5-hydroxymethylcytosine (5hmC), are well known to play critical epigenetic roles in spatially and temporally modulating neuronal gene expression in the mammalian central nervous system (CNS)[1, 2]. A novel DNA adenine modification, N6-methyladenine (6mA), which is prevalent in prokaryotes[3], was recently found in the genome of high eukaryotes, including green algae, worms, fruit flies, frogs, zebrafish, pigs, and mice[4–9]. Although evidence from these studies suggests potential epigenetic roles for 6mA, its precise biological function(s) remain elusive[10]. Furthermore, little is known about the presence and functions of 6mA in the mammalian CNS.

Various genetic factors and environmental cues contribute to mental illness[11]. Among these, exposure to chronic stress is one of the strongest and most direct risk factors for developing numerous psychiatric disorders including depression[12, 13]. Compelling evidence support the fundamental roles of epigenetic alterations in depression as changes in both DNA methylation and histone modifications alter gene expression in brains[14, 15]. The prefrontal cortex (PFC) is highly involved in complex cognitive behavior, but is also sensitive to the adverse effects of stress exposure[16]. In rodent experiments, chronic stress results in dendritic atrophy and spine loss in the PFC[17]. A recent study demonstrated that chronic social defeat stress could decrease DNA methylation levels in PFC[18]. It remains to be determined whether the DNA adenine modification 6mA is present in the mammalian brain and if it is involved in the response to extrinsic challenges such as stress.

Here we examined the dynamic changes of 6mA in mouse PFC in response to stress. We found that global 6mA was significantly elevated upon stress. Genome-wide 6mA and transcriptome profiling revealed an inverse association of 6mA dynamic changes with a set of upregulated neuronal genes or downregulated LINE transposon expression. Genes bearing stress-induced 6mA changes significantly overlap with loci associated with neuropsychiatric disorders. These results together suggest a potential epigenetic role for 6mA in the mammalian brain as well as its possible involvement in mental illness.

## Results

**Chronic stress induces the accumulation of 6mA in mouse PFC.** It is well known that environmental exposure may induce epigenetic changes. To understand the dynamics of 6mA in the mouse brain, we employed a well-established paradigm for chronic stress to model environmental exposure. We exposed 7- to 8-week-old wild-type C57BL/6 adult male mice to normal or restraint environments two hours per day in consecutive 14 days[19, 20] (Supplementary Fig. 1a, b). We then performed the forced swim test (FST) and tail suspension test (TST) to evaluate stress responses in restrained and control mice. There was a significant increase in immobility time in the restrained stressed group in both the FST and TST, indicating that the restraint environment successfully altered mouse behavior (Fig. 1a, b). In order to test 6mA changes in response to stress, we applied the highly sensitive ultra-performance liquid chromatography tandem mass spectrometer (UHPLC-MS/MS) to precisely quantify 6mA dynamic changes upon stress in several brain regions involved in stress response, including the PFC, hippocampus (HIP), amygdala (AMY), and hypothalamus (HYP). Interestingly, although detectable 6mA was present in all of these brain regions, only 6mA in PFC showed substantial and significant increase from an average of 6.6 p.p.m. (6mA per million dA) to 25.5 p.p.m. upon stress (Fig. 1c). Since PFC has been widely linked to critical neuronal activities such as stress response[21, 22], we validated the

6mA dynamic changes in genomic DNA isolated from control and stressed mice PFC using a 6mA-specific antibody, whose specificity to DNA 6mA has been extensively validated in previous publications[4, 5, 7, 9], and by PCR using normal dATP or 6mA-modified dATP (Fig. 1d). Consistent with UHPLC results, 6mA was present in the PFC of wild-type mice and showed ~fourfold significant increase upon stress (Fig. 1d, e and Supplementary Fig. 1c). These data indicate that 6mA is present in mammalian brains and undergoes dynamic changes upon stress.

**Genome-wide 6mA dynamics upon stress exposure.** To understand the potential role of 6mA in stress response, we randomly chose three pairs of control and stressed mice from three independent stress experiments to profile genome-wide 6mA dynamics in PFCs using a published 6mA-immunoprecipitation (IP) protocol coupled with high-throughput sequencing[6, 7, 9]. We ensured the specificity of the 6mA-IP using competitive elution with excess 6mA-modified nucleotides. In order to identify significant differential 6mA dynamic regions upon stress, we applied published computational algorithms edgeR[23] to compare normalized 6mA read density differences between triplicated control and stressed PFC groups across the mouse genome utilizing 500 bp bins based on the 6mA-IP resolution. Differential regions with a $p$-value $< 0.05$ ($-\log10$ $p$-value $> 1.3$, false discovery rate $< 0.077$) were considered significant. We identified 21,974 loss-of-6mA regions and 37,937 gain-of-6mA regions upon stress (Fig. 2a, and Supplementary Data 1, 2). Genomic annotation of these 6mA dynamic regions revealed substantial biased genomic associations. For example, 65% of regions that gained 6mA upon stress were associated with intergenic regions, which is the highest enrichment vs. expected value (Fig. 2b). Intragenic gain-of-6mA upon stress predominantly occurred at introns and was excluded from most coding exons (33%-introns vs. 0.38%-exons, Fig. 2b). The strong association between 6mA and introns and intergenic regions was consistent with observations in mouse kidney[7] and embryonic stem cells[9]. Additionally, 38% of intragenic loss-of-6mA occurred at introns, among the highest enrichment vs. expected value, and 59% of loss-of-6mA were identified at intergenic regions (Fig. 2c). The fact that gain-of-6mA and loss-of-6mA had differential enrichment at intergenic regions or introns vs. expected values may indicate the differential roles of 6mA at these genomic regions. The 6mA reads ratio (stressed vs. control) was plotted across the intragenic regions of genes bearing dynamic 6mA changes upon stress, i.e., red plots (fold change>1) indicate overall gain of 6mA upon stress whereas blue plots (fold change<1) indicate loss of 6mA (Fig. 2d). Interestingly, more 6mA loss was observed around transcription start sites (TSS) than gene bodies in genes bearing loss-of-6mA regions (Fig. 2d). In contrast, gain-of-6mA showed substantially more gene body enrichment than TSS in genes carrying gain-of-6mA regions (Fig. 2d). These observations imply that gain- or loss-of-6mA has differential roles on protein-coding genes.

**6mA negatively correlates with LINE transposon expression.** To further dissect the association between 6mA dynamic changes upon stress and intergenic transposon activities, we analyzed intergenic 6mA dynamic regions at different classes of transposon elements (Fig. 3). We found that 49.2% of intergenic gain-of-6mA was annotated to long interspersed nuclear element (LINE) transposons, which is the most highly enriched transposon over expected value (Fig. 3a). In contrast, only 8.8% of loss-of-6mA regions were located on LINEs, while 33.9% enriched at simple repeats (Fig. 3b). N(6)-methyladenine has been shown to modulate transposon expression in *Drosophila*[6] and target LINE

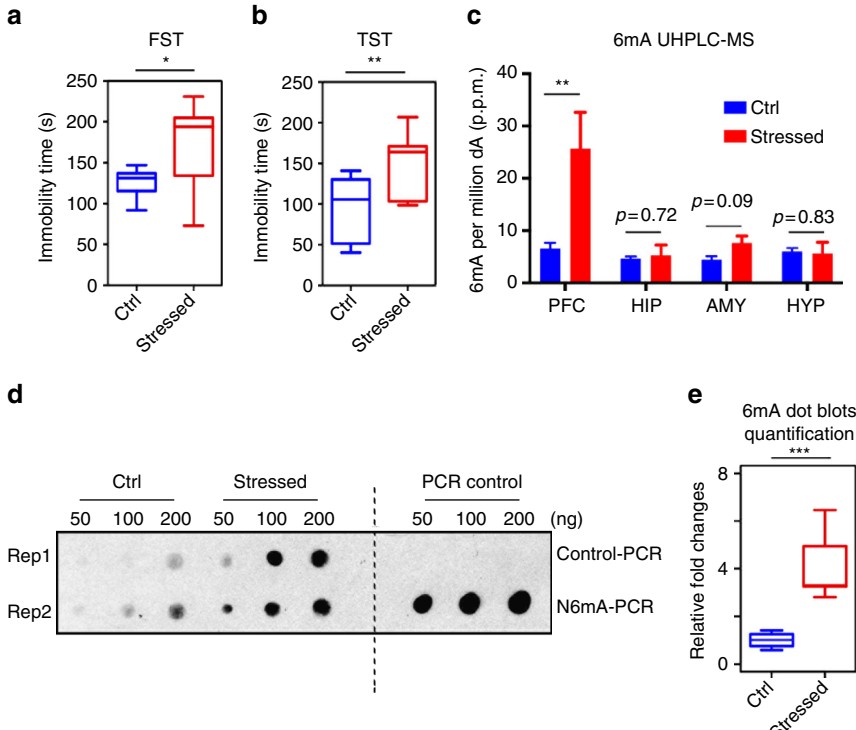

**Fig. 1** Chronic stress induces 6mA accumulation in mouse prefrontal cortex (PFC). **a, b** Chronic stress resulted in increased immobility time in the forced swim test (FST) and tail suspension test (TST). ($n = 9$; $*p < 0.05$; $**p < 0.01$; unpaired $t$-test, error bars = mean±SEM, unpaired $t$-test). **c** Highly sensitive ultra-performance liquid chromatography tandem mass spectrometer (UHPLC-MS/MS) precisely quantified 6mA in prefrontal cortex (PFC), Hippocampus (HIP), Amygdala (AMY), and Hypothalamus (HYP) upon stress. N(6)-methyladenine/total A was indicated as percentage per million dA (p.p.m.). N(6)-methyladenine levels in PFC drastically increased from 6.6 p.p.m. (6mA per million dA) to 25.5 p.p.m. upon stress ($**p < 0.01$; unpaired $t$-test; non-significant $p$-values are indicated). **d** Representative 6mA-specific dot blots revealed accumulation of 6mA in mouse PFC upon stress. Antibody specificity has been validated by detecting PCR products using methylated dATP (6mA-PCR) but not unmodified dATP (control PCR). **e** Quantification of dot blots in **d** by ImageJ software ($n = 4$, $***p < 0.001$; unpaired $t$-test, error bars = mean±SEM)

transposons in mouse embryonic stem cell (mESC)[9]. Stress could also substantially alter transposon expression in mouse PFC (Supplementary Data 3). Expression of transposons, such as LINEs, has been proposed to play important roles in stress response in the mammalian brain[24]. Since 49.2% of intergenic gain-of-6mA upon stress were located on LINEs and highly enriched than expected values (Fig. 3c), which was not found in the loss-of-6mA regions (Fig. 3d), we sought to investigate gain-of-6mA association with LINE expression by RNA-seq. 103 of 120 expressed LINEs in PFC had significant 6mA changes upon stress, with 82 (79.6%) showing increased 6mA. Importantly, the majority (74 out of 82, 90.2%) of LINEs with gain-of-6mA are downregulated upon stress (Fig. 3e). The percentage of down-regulated LINEs bearing increased 6mA was significantly more enriched than observed in short interspersed nuclear element (SINE) transposons ($\chi^2$ test, $p < 0.001$). These data indicate an inverse association between gain-of-6mA and LINE expression upon stress. This is consistent with strong enrichment of gain-of-6mA annotations at LINE elements relative to expected values (Fig. 3a).

To validate the dynamic 6mA regions identified by 6mA-IP-seq, we employed a published restriction enzyme digestion method which takes advantage of the 6mA-sensitive restriction enzyme *Dpn*I that preferentially cleaves methylated adenine at GATC/CATC/GATG sites[25]. Equal amounts of *Dpn*I-digested DNA and undigested control DNA were subjected to quantitative polymerase chain reaction (qPCR) analyses with primers targeting 6mA dynamic regions identified by 6mA-IP. The percentage of 6mA in either control or stressed PFC can be

assessed by qPCR amplification and normalized to undigested DNA control (digested/undigested)[4, 25]. Loci with lower 6mA modification upon stress would hinder *Dpn*I digestion, resulting in higher PCR fold changes than control. We validated 16 random gain- or loss-of-6mA loci annotated to intragenic or intergenic regions identified by 6mA-IP-seq and found consistent 6mA dynamic changes upon stress (Supplementary Figs. 2, 3a, b). Furthermore, we confirmed four additional loss-of-6mA regions annotated in genes involved in neurodevelopment and behavior (Supplementary Fig. 3c). Importantly, no significant difference was found in random loci without 6mA changes, indicating the reproducibility and reliability of 6mA differential region identification (Supplementary Fig. 3d). However, although *Dpn*I can cleave fully methylated adenine at GATC/CATC/CATG to indicate specific methylated adenine in these sites, methylated adenines outside these cutting sites are beyond the scope of *Dpn*I-based 6mA detection, which we could not validate here. Finally, we confirmed differential 6mA regions by an additional computational package (DESeq2)[26], and found substantial and significant overlap between 6mA differential regions identified by DESeq2 and edgeR that was initially used to identify differential 6mA regions. Taken together, these analyses confirm dynamic 6mA regions upon stress identified by 6mA-IP-seq.

**N(6)-methyladenine negatively correlates with a group of neuronal gene expression.** To further link 6mA dynamics and gene expression, global transcriptome changes in PFC upon stress were assessed by RNA-seq (Supplementary Data 4). Gene

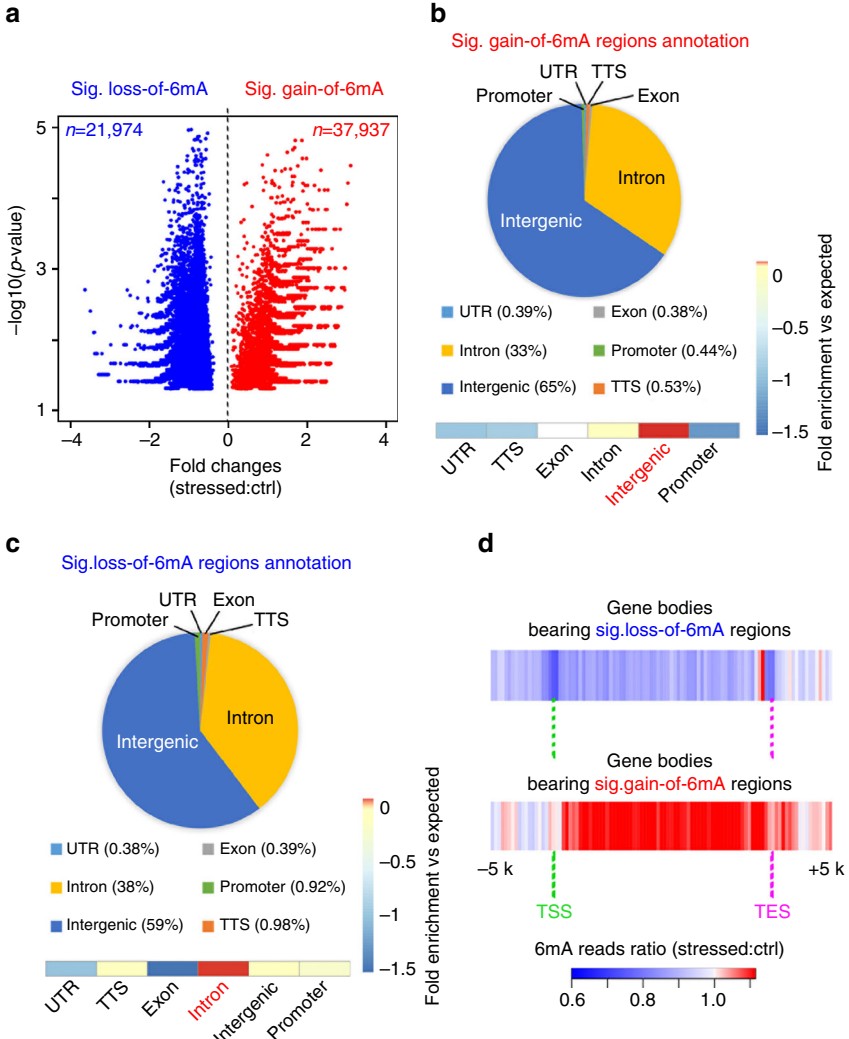

**Fig. 2** Genome-wide 6mA profiling in PFC reveals dynamic changes in 6mA on intragenic regions in response to stress. **a** Volcano plot illustrated the significant gain- and loss-of-6mA regions upon stress. Each dot represents 500-bp binned mouse genome containing significant 6mA reads differences by pair-wise replicate comparison. Fold changes in 6mA reads (Stressed:Control) are indicated on the *x*-axis, and –log10(*p*-value) of each bin are indicated on the *y*-axis. 37,937 significant gain-of-6mA regions and 21,974 loss-of-6mA regions are highlighted in red and blue, respectively. **b, c** Genomic annotation of significant gain- or loss-of-6mA demonstrate their percentage of genomic feature association and enrichment vs. expected values. Dynamic 6mA changes upon stress predominantly occurred on introns and intergenic regions. Gain-of-6mA regions were mostly enriched in intergenic regions over expected value whereas loss-of-6mA enriched in introns over expected values (Heatmap view, highlighted in red). **d** Average fold change between 6mA normalized reads (stressed vs. control) were calculated in gene bodies bearing loss- or gain-of 6mA regions upon stress, respectively, plus 5 kb upstream and downstream flanking regions. Average fold change was plotted in Heatmap view. Red plots (fold change > 1) indicate overall gain of 6mA upon stress whereas blue plots (fold change < 1) indicate loss of 6mA

Ontology (GO) analyses revealed enrichment for numerous biological pathways involved in neuronal functions in the cohort of upregulated genes (Supplementary Fig. 4a). In contrast, these terms were not found in the GO analysis of downregulated genes (Supplementary Fig. 4b), indicating that many critical neuronal genes were upregulated in response to stress. To investigate the potential roles of 6mA in the stress response, we computed the overall 6mA reads ratio at significant intragenic gain- or loss-of-6mA regions per gene between stressed and control PFC, and further subcategorized these genes into four groups: gain-of-6mA/upregulated, gain-of-6mA/downregulated, loss-of-6mA/upregulated, and loss-of-6mA/downregulated genes. Importantly, GO analyses revealed that only upregulated genes bearing significant loss-of-6mA were strongly associated with biological pathways (top 15 terms based on fold enrichment) related to

neuronal development and neuronal functions, such as learning, behavior, neurogenesis, and axon development, which have been shown to play important roles in the brain stress response[27, 28] (Fig. 4a, b and Supplementary Data 5). Cross-examination of the relationship between genes and GO terms suggested that a substantial number of upregulated genes related to behavior were also enriched for other biological functions such as learning and neuron projection development, suggesting that these genes could be involved in multiple biological pathways orchestrating stress response (Fig. 4c). On the other hand, no biological terms related to neuronal functions were found in the other groups of genes (Supplementary Fig. 5a–c). Thus, our findings suggest a specific inverse association between intragenic loss-of-6mA and a set of critical upregulated neuronal genes in response to stress. This data echoes the observation that loss-of-6mA regions were

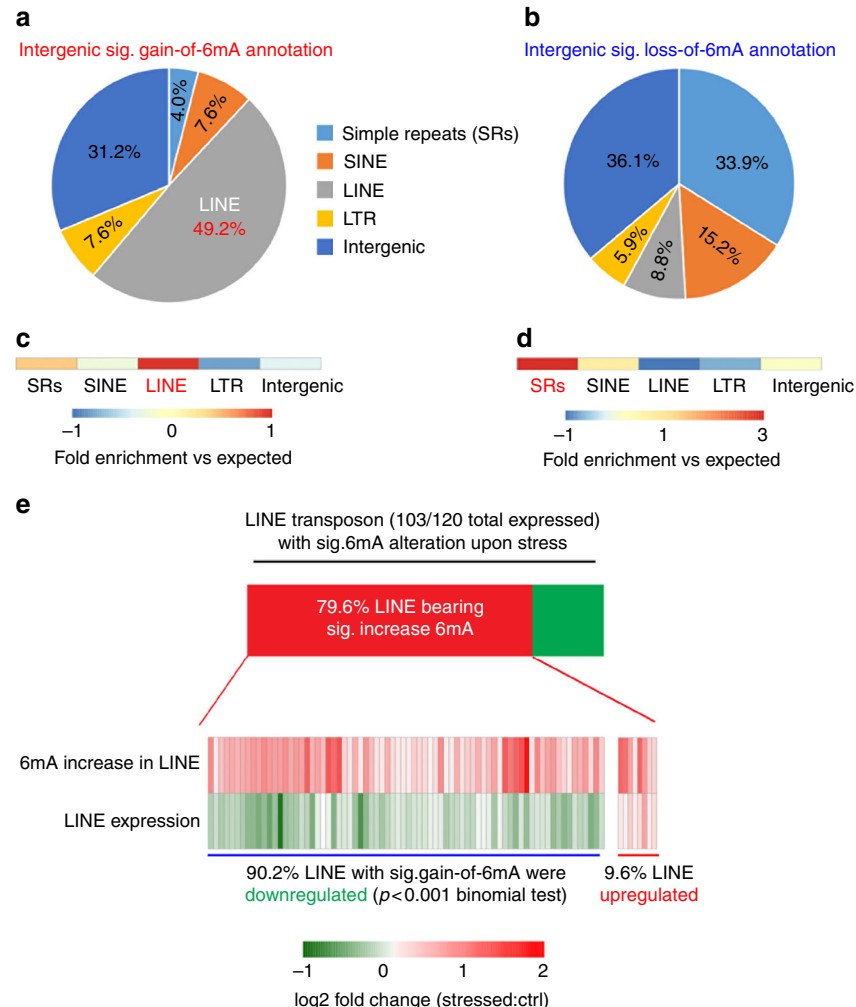

**Fig. 3** Genome-wide 6mA profiling in PFC reveals dynamic changes in 6mA on intergenic repetitive elements in response to stress. **a**, **b** Genomic annotation of significant gain- or loss-of-6mA demonstrate their percentage of repetitive elements association and enrichment vs. expected values. Stress-induced intergenic 6mA alterations occurred on distinct classes of repetitive elements. **c**, **d** The Log2 enrichment of annotated gain- or loss-of-6mA vs. expected values were calculated by HOMER annotation algorithm and demonstrated by heatmap. In total, 49.2% of intergenic gain-of-6mA occurred in the long interspersed nuclear element (LINE), which was highly enriched over expected. Loss-of-6mA was predominantly enriched in simple repeats over expected values. **e** Gain-of-6mA on intergenic regions correlated with downregulation of LINE transposon expression. In total, 79.6% of dynamic 6mA marked LINE transposons possessed gain-of-6mA, with 90.4% of these LINE expression being downregulated upon stress. $P < 0.001$, Chi-squared test comparing to short interspersed nuclear element (SINE) transposon. The fold changes (Stressed:Control) of both 6mA and LINE expression are indicated by Heatmap. Red plots (Log2 fold change > 0) indicate increased expression upon stress (stressed > control), whereas green plots (fold change < 0) indicate decreased expression upon stress (stressed < control)

strongly and uniquely enriched at introns relative to expected values (Fig. 2c).

**Crosstalk between 6mA and other epigenetic mechanisms**. In order to explore the mechanistic roles of 6mA in gene regulation, we investigated the potential correlation between 6mA distribution and enhancer regions and their signature histone modifications[29]. Overall, 6mA levels in both control and stressed PFC samples appeared to be significantly depleted in both general enhancers and cortex-specific enhancers compared to non-enriched input (Supplementary Fig. 6a). Stress further significantly decreased 6mA levels in these regions, implying that altered 6mA upon stress could impact enhancer activity in mouse brains (Supplementary Fig. 6b). H3K27ac and H3K4me1 are associated with enhancers and active promoters[29]. Consistent with enhancer regions, 6mA was also depleted at H3K27ac and H3K4me1[30] regions (Supplementary Fig. 6c). In addition, 6mA

was also depleted at Pol II binding sites, consistent with its negative correlation with transcription. Interestingly, although the same trend (further 6mA depletion upon stress) was observed in H3K27Ac and H3K4me1, stress-induced 6mA depletion was not significant (Supplementary Fig. 6d). This observation suggested that 6mA dynamic changes in response to stress were more specific at *cis*-regulatory regions.

Recent studies suggest that DNA methylation, especially non-canonical CpH methylation (H=A, C or T), plays unique roles in neuronal functions[31, 32]. We explored the correlation between cytosine methylation and adenine methylation in the context of neurons and non-neurons using a genome-wide methylated DNA immunoprecipitation (MeDIP) data set in mouse cortex[33]. Interestingly, methylated cytosine (mC) was strongly enriched at loss-of-6mA regions but only showed modest distributions at gain-of-6mA regions (Supplementary Fig. 7a). The specific enrichment of mC at regions with high 6mA levels in control,

which lose 6mA under stress, could indicate co-existence of these modifications and coordination of repressive roles. Intriguingly, the correlation between mC and 6mA appears to be preferential in neurons, as loss-of-6mA regions have almost two-fold higher mC levels in neurons than non-neuronal cells (Supplementary Fig. 7b). Consistently, global 6mA profiles in control PFC demonstrated a stronger correlation with 5mC than stressed PFC, and the 5mC profile was closer to 6mA profiles in neuronal cells than non-neuronal cells (Supplementary Fig. 8a). Given that a substantial number of gain-of-6mA regions upon stress were annotated in LINE transposons, we tested whether cytosine

methylation was also altered in LINE transposons. Interestingly, MeDIP coupled with qPCR detected a significant increase of methylation in L1td1[34] upon stress (Supplementary Fig. 8b), suggesting potential crosstalk between 6mA and 5mC in LINE transposons. Furthermore, the correlation between cytosine and adenine methylation was supported by the top motifs found in intragenic regions upregulated with loss-of-6mA (Supplementary Fig. 9). "AC" and "CA" are major motifs within these regions in PFC. Moreover, these motifs were predicted to be recognized by various transcription factors, such as Egr2, Foxh1 and Hif2a (Supplementary Fig. 9), suggesting that 6mA dynamics may serve

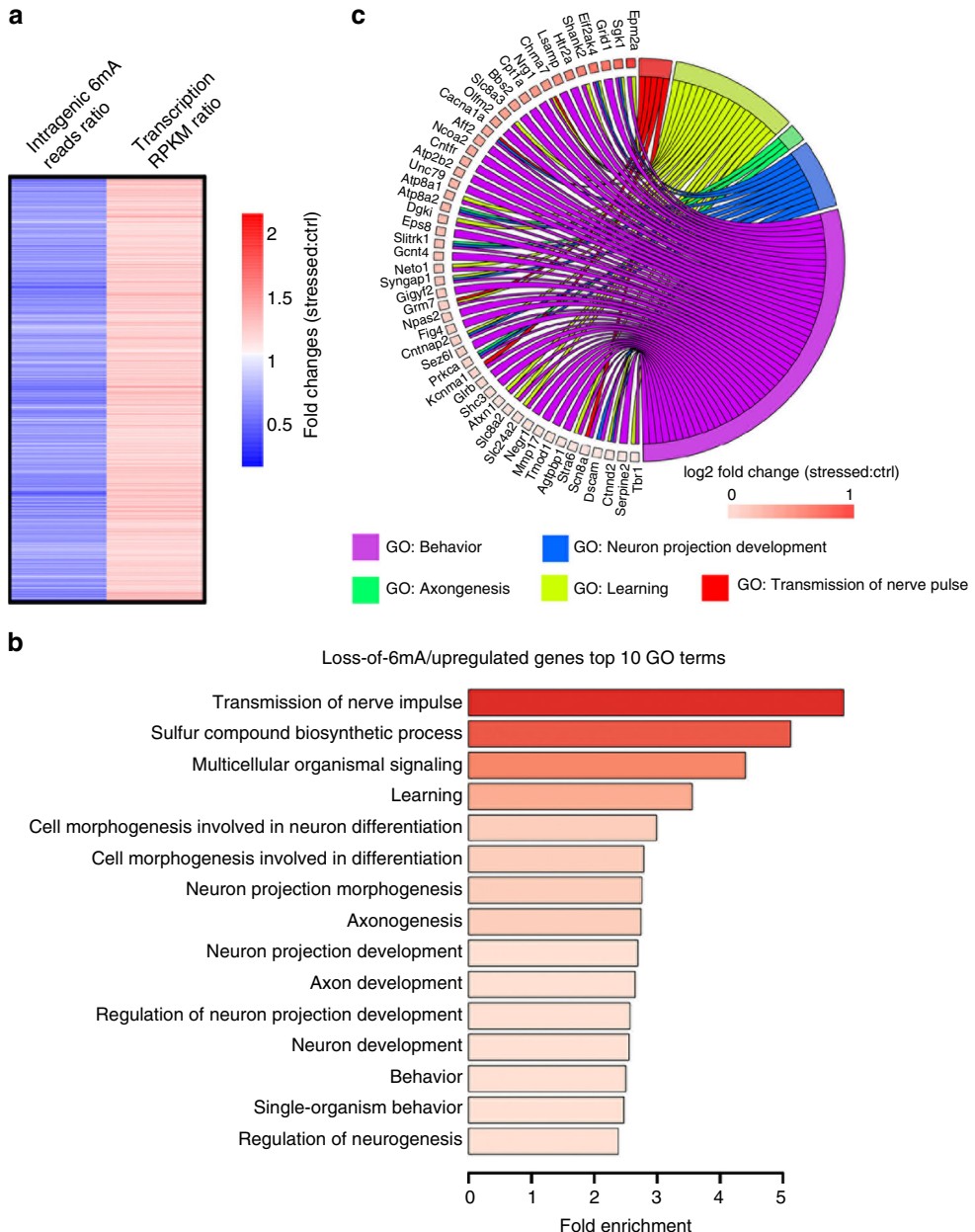

**Fig. 4** *N*(6)-methyladenine negatively correlates with neuronal gene expression in response to stress. **a** Genes with significant loss-of-6mA and increased expression upon stress are indicated. The log2 fold changes (Stressed:Control) of both 6mA and transcription are illustrated by Heatmap. Red plots (fold change > 1) indicate transcription upregulation upon stress, whereas blue plots (fold change < 1) indicate loss-of-6mA on the same genes upon stress. **b** Genes in **a** were enriched in pathways related to neuronal functions, neurogenesis and behavior control. Fold enrichment of each GO term are indicated by the *x*-axis and bar color. **c** Circos plot to indicate the relationship between genes and GO terms. Cross-examination of GO analyses suggested that a substantial number of upregulated genes with loss-of-6mA related to behavior were also enriched in other biological functions such as learning and neuron projection development. Log2 fold changes of gene expression were indicated as colored squares

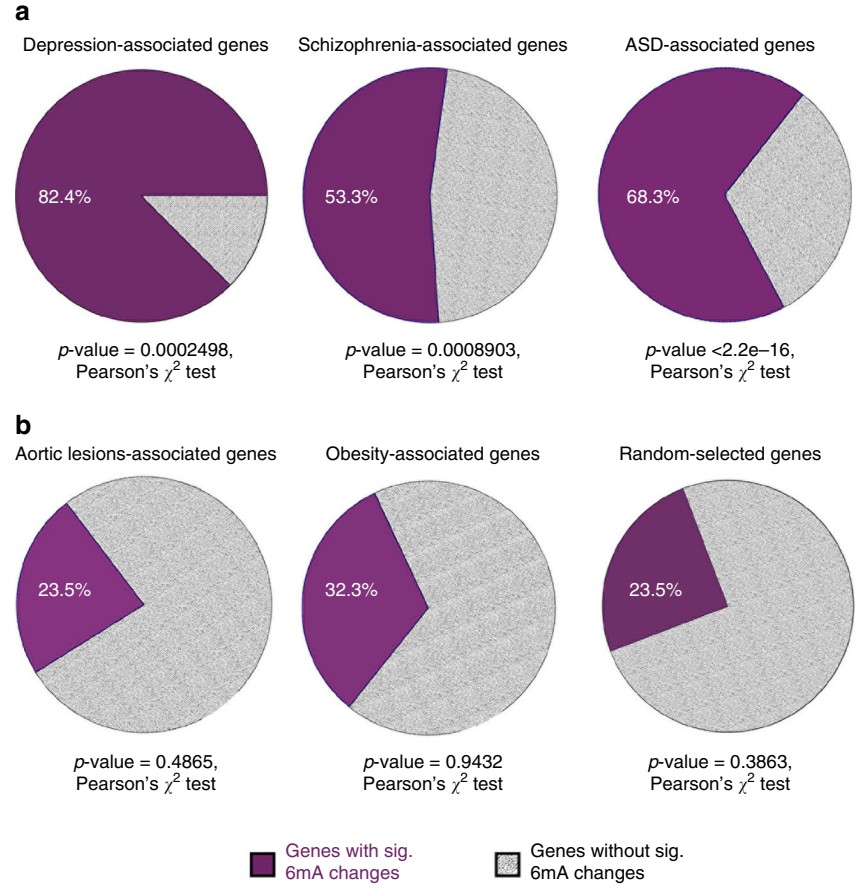

**Fig. 5** Dynamic 6mA is associated with depression-linked genetic loci. **a** Significant overlap between depression-associated genes containing the top 17 most significant SNPs, schizophrenia (SCZ)-related risk genes and autism spectrum disorders (ASD)-related risk genes with dynamic 6mA marked genes upon stress are indicated. *P*-value was calculated by Pearson's $\chi^2$ test. **b** The genes related to aortic lesions, obesity or randomly selected genes did not show significant enrichment. Statistical significance was calculated by Pearson's $\chi^2$ tests, and *p*-values are indicated

as docking sites to modulate transcription factor binding in response to stress. In fact, 6mA is enriched in histone modifier binding sites, such as Polycomb PRC1 component BMI1 Proto-Oncogene, Polycomb Ring Finger (Bmi1), but depleted in another Polycomb protein RING1B (Supplementary Fig. 10a), implying potential and specific crosstalk between 6mA and other epigenetic mechanisms. Significant loss-of-6mA in stress was found at Bmi1-binding sites (Supplementary Fig. 10b), and Bmi1 has been shown to mark certain enhancer regions[35]. This observation was consistent with the depletion of 6mA from enhancers in normal mouse PFC (Supplementary Fig. 6).

**Dynamic 6mA is associated with depression-linked loci.** To shed light on the functional relevance of 6mA dynamics in the stress response and human neuropsychiatric disorders, we obtained human depression-associated genes related to the 17 most significant SNPs[36], and correlated their mouse orthologs with 6mA dynamic genes upon stress in our study. Interestingly, 14 out of 17 (82.4%) depression-associated genes significantly overlapped with 6mA dynamic genes upon stress (Fig. 5a, Pearson's $\chi^2$ tests). Importantly, the same number of random genes selected to match the GC content, gene size, and neuronal expression of these depression-associated genes did not significantly overlap with 6mA dynamic genes upon stress (Supplementary Fig. 11). To further investigate whether 6mA is associated with other neuropsychiatric disorders, we obtained genes associated with autism spectrum disorder (ASD)[37], schizophrenia

(SCZ)[38], aortic lesion, and obesity[39]. N(6)-methyladenine dynamic genes showed significant and specific overlap with ASD and SCZ genes (Fig. 5a, Pearson's $\chi^2$ test), but not with genes involved in aortic lesion, obesity[39], or an equal number of randomly selected refseq genes (Fig. 5b, Pearson's $\chi^2$ test). Together, these data suggest a potential link between this novel DNA modification and mental illness. Furthermore, genes with the top 10,000 depression-associated SNPs also displayed significant overlap with 6mA dynamic genes but not with an equal number of random genes (Supplementary Fig. 12a, Pearson's $\chi^2$ tests). We generated three random gene lists to match the GC content, gene size, and neuronal expression to the full list of depression-associated genes, and calculated their overlap with 6mA dynamic genes in stress (Supplementary Fig. 12b–d). Similar to the top 17 most significant SNPs, the overlap between 6mA dynamic genes in stress and the full list of depression-associated genes is also specific and significant, as all three random gene lists did not show significant overlap with 6mA dynamic genes (Supplementary Fig. 12b–d, Pearson's $\chi^2$ tests). These data indicate a strong association between 6mA and loci linked to neuropsychiatric disorders.

**Discussion**

N(6)-methyladenine is relatively well characterized in bacteria and is involved in host defense against bacteriophages and transposons, modulation of DNA and chromosome replication, DNA mismatch repair and expression of certain operons[3]. The presence of 6mA in the genome of eukaryotic cells, including

green algae, worms, and fruit flies, was recently documented[4–6]. Our data presented here for the first time demonstrate the presence and dynamics of 6mA in the mouse brain in response to stress. Genome-wide 6mA profiling indicates an inverse association of 6mA dynamics with a group of neuronal genes involved in the stress response as well as transposon LINE expression. *N*(6)-methyladenine appears to be depleted in *cis*-regulatory elements and correlates with cytosine methylation and other epigenetic regulation in the mammalian CNS. Interestingly, the group of genes in mouse PFC bearing dynamic changes in 6mA in response to stress significantly and specifically associate with genes involved in neuropsychiatric disorders.

Increasing evidence indicate that physiological and psychological stress could change DNA methylation at critical genes related to stress in mouse brains[18, 40, 41]. Furthermore, DNA methylation alterations are further supported by similar observations in postmortem human brains of patients diagnosed with major depressive disorder who committed suicide[42]. PFC is responsible for highest-order cognitive abilities and is particularly susceptible to chronic stress and plays significant roles in depression[22]. Our results demonstrate specific accumulation of 6mA in PFC under chronic stress, which further supports the critical roles of PFC in the stress response and suggest the potential importance of 6mA dynamics during this process. Interestingly, it has been suggested that chronic stress could induce morphological and molecular changes within PFC to render cognitive rigidity and increased vigilance[28]. These changes are PFC specific, and could be conveyed by the rapid alteration of 6mA upon stress. It is important to note that the stress condition applied in our current study only represents one of many unknown environmental paradigms which could spatially and temporally trigger 6mA dynamics to deal with environmental changes at the epigenetic level. Thus, further studies will be important to precisely elucidate the roles of 6mA in specific brain regions in response to various stimuli.

Earlier experiments have found that transfecting a gene with 6mA modifications could result in an almost 30-fold decrease in transcription factor binding in mammalian cells, implying potential roles for 6mA in transcription[43]. In the current study, we show that significant loss of intragenic 6mA in mouse PFC upon stress inversely correlates with a group of upregulated genes enriched in neuronal functions, neurodevelopment and behavior. These data indicate that the presence of 6mA on these genes could play a negative transcription role under normal conditions in mouse PFC, consistent with a recent study in mESCs[9]. Interestingly our data suggest a general depletion of 6mA in enhancer regions and Pol II binding sites (Supplementary Fig. 5c), indicating that 6mA repression could be in part due to its modulation of intronic *cis*-regulatory regions. Genome-wide 6mA distribution appears to be tissue-specific, as dynamic 6mA changes are highly enriched in "CA" motifs in mouse PFC compared to "GA" motifs in static mouse kidney[7]. Increasing evidence supports that non-CpG methylation (mCH, H=A/C/T) more directly regulates genes in neurons, with mC in the "CA" context as a major motif[32]. It is important to further understand the precise and tissue-specific roles of 6mA and investigate the potential link between cytosine and adenine methylation.

Our finding that the group of genes bearing dynamic 6mA changes in response to stress significantly and specifically overlaps with genes involved in depression, SCZ, and ASD is particularly intriguing. Potential 6mA "readers", "writers", and "erasers" that maintain 6mA homeostasis in the CNS could play key roles in maintaining normal brain function. It will be important to identify the 6mA functional players in future studies. Thus, aberrant 6mA in response to stress could contribute to neuropsychiatric diseases by ectopically recruiting transcription

factors and altering target expression. Therefore, our finding provides the first link between this novel DNA adenine methylation and neuropsychiatric disorders.

## Methods

**Animals.** All wild-type mice had the C57BL/6 genetic background. All mice used in this study were male and 7–8 weeks old. All animal procedures were performed according to protocols approved by Institutional Animal Care and Use Committee at Emory University.

**Chronic restraint stress.** Mice were individually placed into a customized, well-ventilated 50-ml conical tube (Corning Inc., Corning, NY) daily from 10 a.m. to 12 p.m. for 2 h in their home cages for 2 weeks. The restrained mice could move from the supine to prone position, but were not able to move forward or backward. The non-restrained control mice remained undisturbed in their home cages. After restraint stress, mice were released from the tube and returned to their home cages. Three independent stress experiments were performed with nine animals each in control and stressed groups per experiment.

**Forced swim test.** The FST was performed as described previously[44]. In brief, each mouse was placed in a glass cylinder (20 cm high, 15 cm in diameter) with warm water (23–25 °C, 14 cm in depth) for 6 min. The water in the glass cylinder was replaced with fresh water of the same temperature for each individual test. Mice were gently dried after removal from the bath and returned to their home cages. Immobility time was recorded when the animal was floating in the water or making the minimum movement necessary to float in the water. An unpaired *t*-test was used and error bars represente mean±SEM. Investigators were blinded to groups and genotypes.

**Tail suspension test.** The TST was performed according to the method outlined in previous reports with a minor modification[45]. Each mouse was suspended by adhesive tape placed ~1 cm from the tip of the tail. The distance between the tip of the tail of each mouse and the desktop was about 45 cm. Each mouse was suspended for 6 min, and the immobility time was recorded. Unpaired *t*-test was used and error bars represent mean±SEM. Investigators were blinded to groups and genotypes.

**Genomic DNA and RNA isolation.** Mouse prefrontal cortexes were dissected from the brain immediately after the behavior tests. Genomic DNA was isolated from different brain tissues in 600 µl of digestion buffer (100 mM Tris-HCl, pH 8.5, 5 mM EDTA, 0.2% SDS, 200 mM NaCl), then treated with Proteinase K (Thermo) at 55 °C overnight. The second day, 600 µl of phenol:chloroform:isoamyl alcohol (25:24:1 saturated with 10 mM Tris, pH 8.0, 1 mM EDTA) (P-3803, Sigma) was added to samples, mixed completely, and centrifuged for 10 min at 12,000 rpm. The aqueous layer was transferred into a new Eppendorf tube and precipitated with 600 µl of isopropanol. The pellet was washed with 75% ethanol, air-dried, and resuspended with Nuclease-Free Water (Ambion). DNA samples were treated extensively with an RNase cocktail (Ambion) to remove contaminating RNA. PFC samples were homogenized in Trizol (Invitrogen) and processed according to the manufacturer's instructions for total RNA isolation.

***N*(6)-methyladenine antibody specificity.** The specificity of 6mA antibody (Synaptic systems) was extensively validated to ensure that it did not cross-react with 5mC, 3mA, or denatured 1mA[5]. It displayed poor reactivity with RNA molecules[5]. In addition, consistent with our data, a recent publication[9] used this LC–MS approach to analyze DNA modifications in mouse embryonic stem cells. No 1-methyladenine (1mA), 3-methyladenine (3mA), or 3-methyl-cytosine (3mC) was detected in the mouse genome. The specificity of this antibody was validated by PCR using normal dATP or 6mA-modified dATP (Fig. 1d).

***N*(6)-methyladenine dot blot.** Dot blot was performed on a Bio-Dot Apparatus (#170-6545, BIO-RAD) as described previously[46] using a specific 6mA antibody (Synaptic Systems, cat. no. 202 003) incubating overnight at 4 °C. Horseradish peroxidase-conjugated antibody to rabbit (1:5000, #A-0545, Sigma) was used as a secondary antibody, and incubated for 45 min at 20–25 °C. Standard DNA templates were loaded (D5405, ZYMO) for quantification and to verify antibody specificity. The density of each dot signal was quantified by ImageJ software[47]. An unpaired *t*-test was used and error bars represent mean±SEM. Extensive RNase treatments were performed to ensure complete removal of RNA contamination.

**UHPLC-MS/MS analysis.** Genomic DNA of PFC, HIP, AMY, and HYP were enzymatically digested into single nucleosides with a mixture of DNaseI, calf intestinal phosphatase, and snake venom phosphodiesterase I at 37 °C for 12 h. After enzymes were removed by ultrafiltration, the digested DNA was subjected to UHPLC-MS/MS analysis. HPLC fractionation of mouse m6dA and UHPLC-QTOF-MS/MS analysis were performed as described previously[6]. The 6mA levels

in control PFC (6.6 p.p.m.) and stressed PFC (25.5 p.p.m.) had similar ranges as observed in mouse embryonic stem cells[9].

**N(6)-methyladenine immunoprecipitation.** Genomic DNA was sonicated to 200–300-bp fragments for IP-based 6mA enrichment using rabbit polyclonal antibody raised against 6mA (Synaptic Systems) at 1:100 in 1× IP buffer containing 100 Mm Tris-HCl pH 7.4, 150 Mm NaCl, 0.05% Triton X-100. The IP was conducted with overnight platform rotation at 4 °C without beads. Dynabeads Protein G (Novex by Life Technologies, REF 10009D 30 mg/ml) were added, and the mixture was rotated at 4 °C for 2 h. Beads were washed six times at room temperature with 1× ice-cold IP buffer, 5 min each with rotation. After the wash, the DNA fragments were eluted by either proteinase K digestion or 6mA competition. For proteinase K digestion-based elution, the beads were treated with digestion buffer containing 1× TE, 0.25% SDS, and 0.25% Proteinase K (2.5 mg/ml) at 55 °C and centrifuged at 1400 rpm for 2 h. The eluted DNA fragments were extracted once with phenol:chloroform:isoamyl alcohol=25:24:1, followed by extraction with chloroform alone, and then precipitated with glycogen, NaOAc pH 5.2, and 100% ethanol at −20 °C for at least 2 h or overnight. For 6mA base competition-based elution, the beads were treated with 1× IP buffer with 2.6 mM 6mA three times for 30 min each at room temperature. The eluted DNA fragments were precipitated without extraction with phenol and chloroform.

**N(6)-methyladenine DpnI digestion and qPCR.** DpnI digestion and qPCR were conducted as previously described[25]. Briefly, restriction enzyme digestion was performed by treating 1 μg of genomic DNA with 5 μl of 5 U/μl DpnI restriction enzyme (NEB) at 37 °C for 1 h. The digested DNA and non-digested DNA (5 ng) were subjected to qPCR using FastStart SYBR Green Master kit.

**Library preparation and high-throughput sequencing.** Enriched DNA from triplicate 6mA-IP were subjected to library construction using the NEBNext ChIP-Seq Library Prep Reagent Set from Illumina according to the manufacturer's protocol. Briefly, 25 ng of input genomic DNA or experimental enriched DNA were used for each library construction. DNA fragments (150–300 bp) were selected by AMPure XP Beads (Beckman Coulter) after adapter ligation. An Agilent 2100 BioAnalyzer was used to quantify amplified DNA, and qPCR was applied to accurately quantify library concentration. In total, 20 pM diluted libraries were used for sequencing. Fifty cycle single-end sequencing reactions were performed using Illumina HiSeq 2000. Image processing and sequence extraction were done using the standard Illumina Pipeline.

RNA-seq libraries were generated from triplicate samples per condition using the Illumina TruSeq RNA Sample Preparation Kit v2 following the manufacturer's protocol. RNA-seq libraries were sequenced as 50-cycle paired-end runs using Illumina HiSeq 2000.

**Sample size and statistical analyses.** Eighteen wild-type mice were randomly divided into control and stressed groups. No particular animals were included or excluded from the study. For 6mA profiling, three biological replicates (both control and stressed mouse PFC DNA) from different batches of stress experiments with consistent PFC 6mA increases were subjected to 6mA-IP-seq. RNA-seq was also performed on three biological replicates. P values from Pearson's $\chi^2$ tests and t-tests are indicated in Figures and Figure legends. All statistical analyses were performed in the R computational environment.

**Bioinformatics analyses.** Triplicated FASTQ sequence files from control and stressed 6mA-IP in PFC were aligned to the mm9 reference genome using Bowtie v1.1.2[48]. Established computational algorithms (edgeR package in R/Bioconductor environment) using quantile-adjusted conditional maximum likelihood methods were used to compare three control and three stressed normalized 6mA read density in 500 bp binned mouse genome (6mA-seq resolution) with the consideration of sample variation. Differential regions with a p value<0.05 (false discovery rate<0.077) were considered significant. The differential 6mA regions were further validated by DESeq2, another widely utilized computational algorithm[26]. Ngs.plot software was used to calculate 6mA and 5mC concatenated reads on various genomic regions to plot average 6mA reads on gene bodies[49]. Annotation analyses and expected values of genomic features were performed using the HOMER suite[50]. Motif search and transcription factor prediction were performed by regulatory sequence analysis tools[51] or HOMER suite. RNA-seq reads were aligned using TopHat v2.0.13[52], and differential reads per kilobase of transcript per million mapped reads (RPKM) expression values were extracted using Cuffdiff v2.2.1[52]. Genes with RPKM values more than 0.5 and log2 fold changes more than 0.1 were kept for transcriptome analyses. All differentially expressed genes were tested for correlation to 6mA dynamic changes since stress could result in subtle changes in expression. Gene ontology analyses were performed by Gene Ontology Consortium[53]. Transposon expression was analyzed by TEtranscripts[54]. Circos plot was generated by R-package GOplot[55]. The global Spearman correlations between 6mA and 5mC were performed by deepTools[56].

**Published datasets.** Cortex-specific enhancer and cortex general enhancers were obtained from the ENCODE project[29] (http://chromosome.sdsc.edu/mouse/download.html). Enhancer regions were determined as 250 bp flanking the summit. Eight-week-old mouse cortex H3K27ac and h3K4me1 ChIP-seq data were obtained from GSE49847[29]. Mouse anterior cingulate cortex (ACC) DNA methylation (MeDIP-seq), Histone H3K27ac, and H3K4me1 ChIP-seq data were obtained from GSE74965 and GSE74964, respectively[33]. Mouse brain AUTS2, Bmi1, and RING1B ChIP-seq data were obtained from GSE60409[30]. The human loci related to depression, SCZ, and ASD were converted to their mouse orthologs and overlapped with loci bearing dynamic 6mA changes upon stress. All SRA files were downloaded, converted to FASTQ files and processed as described in the bioinformatics methods section.

**Data availability.** The sequencing data have been deposited at Gene Expression Omnibus under accession number GSE79543. Other data showing the significant loss- and gain-of-6mA; stress-induced transposon expression; stress-induced gene expression alteration and upregulated genes bearing significant loss-of-6mA are shown in Supplementary Data 1–5. All other relevant data are available from the corresponding author.

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

## Acknowledgements

We would like to thank B.L. Philips for critical reading of the manuscript. This work was supported in part by NIH grants (NS051630, NS079625, MH102690 and NS097206 to P.J.)

## Author contributions

B.Y., Y.C. and P.J. conceived and designed the project. B.Y., Y.C., Z.W., Y.L., L.H., W.Z. and D.C. performed the experiments. B.Y. performed the bioinformatics analyses. L.C. assisted with the computational coding. D.C. and B.T. contributed the reagents. B.Y. and P.J. wrote the manuscript. All authors commented on the manuscript.

## Additional information

**Competing interests:** The authors declare no competing financial interests.

