## [Peer Review File · Nature Communications]

Reviewers' comments:

Reviewer #1 (Remarks to the Author):

This is an intriguing report on a series of observations regarding the existence of and regulation of N6-methyladenine epigenetic modifications on DNA in the mammalian brain. The N6mA was quantified by LC/MS and PCR-based dot blot, as well as immunoprecipitated. A several fold increase was detected specifically in the prefrontal cortex of mice subject to restraint stress. Genomic analysis suggests that N6mA is selectively enriched in the intergenic regions and even more selectively in LINES in animals exposed to stress. In contrast, N6mA is reduced in intragenic regions following stress and GO analyses suggests stress-induced up regulation of gene expression associated with neural functioning. The importance of enhancers in N6mA regulation of gene expression is implied via comparison to published data on methyl cytosine mapping and various histone modifications. Finally, potential clinical relevance was implied by comparison to published data sets of SNPs and gene sets relevant to neuropsychiatric disorders and two non-nervous system disorders. A strong association with Depression and weaker associations with schizophrenia and autism genes are observed.

The strength of this report is the novel focus on N6mA which is being increasingly recognized as an important epigenetic modifier in other species and mammalian tissues. A series of observations of both currently generated data and mining of published studies are strung together to make a coherent picture of stress induced changes in brain epigenetics and a possible relation to the risk for depression. The weakness of the study is that it consists of a series of observations strung together. There is no causal evidence that N6mA changes are associated with risk of depression or other neuropsychiatric disorders. As such, the conclusions of the study exceed the data. Nonetheless, this is a very intriguing set of observations.

Specific Comments:

1) Pg 6 – the authors state – “To understand the potential role of N6mA in stress response, we randomly chose 3 pairs of control and stressed mice from different batches of stress experiments to profile genome-wide N6mA dynamics in PFCs using published N6mA-IP coupled with high-throughput sequencing” – and then cite three studies, one in drosophila, one that is mostly in xenopus and one in embryonic stem cells. Thus the beginning and end of this sentence are difficult to reconcile. What does “different batches of stress experiments” mean? And how could published data from drosophila, xenopus and stem cells be used to “profile genome-wide N6mA dynamics in PFCs”? In subsequent sentences it becomes clear the authors conducted their own genome wide profile following IP with N6mA antibody, but this is more difficult for the reader to discern than need be.

2) Pg 8 – statements about GO analyses here are also confusing as well as vague. What does it mean to say “critical neuronal genes were upregulated in response to stress”. What makes a gene critical? And isn't it self evident that the same genes would not be down regulated by stress that were upregulated?

3) Pg 10 – what does it mean to say – “Consistent with enhancer regions, N6mA was also

depleted at H3K27ac, H3K4me1, and P300 regions (Supplementary Fig. 5c). However, no significant N6mA changes were observed in these regions” – these two statements seem contradictory.

4) Figure 1 – there is no need to give precise values within the legend. The dot blots are PCR based but is this the most stringent means for quantifying a difference? Why not qPCR? The results would have been strengthened by including additional brain regions here to confirm effects are specific to prefrontal cortex.

5) Figure 2 – a) the volcano plot looks strange in that there appears to be a cut-off for the log₁₀p value as well as some restriction that results in a non-continuous distribution. Is this a by-product of the pair-wise replicates?

6) Figure 2 – b and c) the heat map of fold change here is also confusing. Is this just replicating what is in the pie charts? How is a fold enrichment of zero the most important?

7) Figure 3 creates similar confusion, especially with the pie chart and the heat map using different color coding

8) Figure 4 – b) It is hard to gloss over the observation that “sulfur compound biosynthetic process” is almost as enriched as those related to “transmission of nerve pulse”. The use of the term “nerve pulse” is also unusual, presumably the authors are referring to the action potential?

9) Lastly, the paper needs judicious editing for English.

Reviewer #2 (Remarks to the Author):

In this manuscript, Yao et. al, demonstrated the dynamic N6mA level change in mouse brain upon environmental stress. The authors convincingly showed the presence N6mA in mammalian brain and also offered hints to its functions. These novel findings provide important clues to elucidate the functions of N6mA, a new mammalian DNA modification. In particular, the authors found that overall N6mA level was significantly elevated upon stress, and confirmed the transcriptional repression function of N6mA with high-throughput sequencing data. With those results, this work implied the role of N6mA in mammalian brain in epigenetic silencing as well as its potential involvement in neuropsychiatric disorders. Although authors’ data seem to support the main conclusions in the paper in general, there are still several major concerns that need to be addressed before this manuscript being considered for publication

Major points:

1. The major conclusions were primarily based on genomic sequencing data. Although N6mA-IP-Seq is indeed a reliable approach for detecting genomic N6mA distribution, orthogonal approach is also needed to validate the results. Authors used N6mA-sensitive restriction enzyme DpnI with qPCR, but restriction sites of this enzyme are enriched in *E. coli* genome, but not in mouse. Thus, it is unclear which conclusions can be drawn from this approach. For example, authors can confirm their conclusions with SMRT sequencing, which can detect N6mA directly.

3. For the N6mA bioinformatics analysis, authors used “edgeR” based method to get differential deposition regions. Although this is reasonable, the data need to be analyzed by other well-known statistic models which interrogate differential deposition. Otherwise it will

be difficult to evaluate the robustness of their conclusions.

4. The authors mentioned “only 8.8% of loss-of-N6mA regions located on LINEs, while 33.9% enriched on simple repeats” upon stress. In general, quantification at simple repeats are always problematic. Different statistical models may render different results, which reinforces the necessity to analyze the data thoroughly (see above).

6. Interestingly, authors found that methylated cytosine (mC) was strongly enriched at loss-of-N6mA regions but only showed modest distributions at gain-of-N6mA regions. This could offer an important mechanistic link of N6mA and 5mC. How is the 5mC level of LINE1s upon stress?

Minor points:

1. Why does only PFC show substantial N6mA increase with stress? This seems to be a very interesting observation. The authors could offer some hints or explanation in discussion.

2. The authors found that 49.2% of intergenic gain-of-N6mA was annotated to long interspersed nuclear element (LINE) transposons, which is the highest enriched transposon over expected. However, the authors need to elaborate on their approach for mapping transposons to the genome: how are they aligned and counted? Using unique mapping reads or assign the reads randomly.

Reviewer #3 (Remarks to the Author):

The study by Yao et al investigates the N6-methyladenine modification (N6mA) on DNA in the mammalian brain. While N6mA has been well characterized in prokaryotes its existence in multicellular eukaryotes has only been suggested recently. However its functional importance is still largely unknown although a potential connection in transposon control has been recently described in *Drosophila* and mouse embryonic stem cells.

Here the authors investigated the level of N6mA in mouse brains, in control and stress-induced conditions. The authors found a prominent increase in N6mA levels upon stress, which largely results from enrichment at transposons, specifically on LINE elements. These changes significantly correlate with decrease in transposon levels, thus confirming the demonstrated negative role of N6mA on transposon expression in embryonic stem cells. In contrast decrease in N6mA levels was also observed but in this case these changes occur prominently at intragenic regions, in particular within introns. The authors went on to correlate these specific changes with additional features (gene expression, enhancers, gene class). They found that loss of N6mA is significantly associated with genes involved in neuronal functions.

This work adds novel insights into the N6mA DNA modification in vertebrates, in particular in brain function and therefore should be of interest for Epigenetic community. The paper is in general well written and clear. However the paper suffers from two major limitations. On one hand the majority of enrichments are rather small and it is not clear how the authors filter the noise and how they decide for a particular cut-off to call a peak significant.

Second, if we admit that the peaks are real the study remains largely correlative and the

mechanistic insights gain is rather limited. Despite these hurdles the correlative findings are interesting and suggest an important role for N6mA during stress conditions. For this reason if the authors can clearly justify how they call significant peaks and rule out potential contamination from symbionts the work remains of interest and to my opinion should deserve publication in Nature communications.

Major

As shown in previous studies the reported level of N6mA is very low, near the limit of the detection level (0.0006% of total adenines in PFC). Given the fact that some of these methylated adenines can originate from contamination from microbiota any observed changes on N6mA level must be called with extreme caution. As said above the majority of the enrichments, which are presented as different, looks rather weak. The authors should tell what cut-off they used and explain why this arbitrary cut-off is strong enough to filter the noise. It might be that majority of differences will be filtered out.

As I understood the phrase "significant enrichments" refers to the identified N6mA peaks over the input sample all originating from the same animal. This was done for all animals studied, and then these significant peaks were compared to the corresponding ones but originated from different animals. The resulting fold changes are illustrated on figs 2a and d.

1. What were the parameters used for the identification of the peaks to call them "significant enrichment"? The authors used an independent method by restriction enzyme digestion to validate few of these changes. This should be extended to at least a dozen in total and should comprise genes that show the highest N6mA changes between normal and stress conditions as well as the lowest (to validate their cut-off). I doubt that all the called peaks would be positive in this case.

2. When these "significant peaks" were compared between the treated and non treated samples, what was the fold change cut-off used to claim that regions are enriched or depleted of N6mA. From the fig 2a I can assume that the majority of the regions were enriched or de-enriched with a fold difference of 1.5 or -1.5 (x axis) and only a few regions had a fold enrichment or de-enrichment of >2. Most of the peaks in fig 2d have a scale ranging from 0.6 to 1.2 and it is hard to believe that an enrichment of 1.2 is a real enrichment, especially when the basal level is so low. Please comment on this and justify why the 1.2 enrichment are claimed to be significant. Same problems I see for the de-enrichment in supplementary figs 5 and 6.

3. In supplementary figs 5 and 6 the blue color gradient is used very extensively to show the de-enrichment. However this is misleading. When the color-coding is compared to the actual scale bar it is hardly distinguishable from the 1x fold change, suggesting that barely anything is happening. I suggest using either proper scale for a blue color with a maximum corresponding to the red color, or just avoid using it at all.

4. The authors showed a correlation between m5C and changes in m6A levels upon stress. It would also be informative to have an overall correlation between m5C and m6A in the

brain in wild type condition (as well as upon stress if possible).

5. What is the exact association between N6mA changes upon stress and the genes that are linked to neuropsychiatric disorders shown in fig. 5? Do these genes show loss of m6A and are up-regulated, as shown before for the neuronal genes, or is this association more random, which might suggest some indirect effects.

Minor points:

1. In fig 3c the color-coded scale showing the fold change is logarithmic, but the range is relatively small. For better clarity and consistency with other figures it would be better to keep it linear.
2. In supplementary fig 5c the authors examined Pol II enrichment (and not P300 as mentioned in the text).
3. P11: figure 5a (not 6a)

Responses to Reviewers' Comments:

We would like to thank the reviewers for their thoughtful and constructive comments. Following their suggestions, we have carried out substantial amount of new experiments to address the concerns they raised. These additional experiments confirmed our original conclusions and significantly strengthened this manuscript. Below are the point-by-point responses to each reviewer.

Reviewer #1

1. *“Pg 6 – the authors state – “To understand the potential role of N6mA in stress response, we randomly chose 3 pairs of control and stressed mice from different batches of stress experiments to profile genome-wide N6mA dynamics in PFCs using published N6mA-IP coupled with high-throughput sequencing” – and then cite three studies, one in drosophila, one that is mostly in xenopus and one in embryonic stem cells. Thus the beginning and end of this sentence are difficult to reconcile. What does “different batches of stress experiments” mean? And how could published data from drosophila, xenopus and stem cells be used to “profile genome-wide N6mA dynamics in PFCs”? In subsequent sentences it becomes clear the authors conducted their own genome wide profile following IP with N6mA antibody, but this is more difficult for the reader to discern than need be.”*

We thank the reviewer for pointing this out and we apologize for the confusion. As described in the methods, we have performed three independent restraint stress experiments, with each experiment containing 9 pairs of age-matched littermates in either control or stressed groups. Three pairs of control or stressed animals were randomly chosen from these three independent experiments (described as “different batches of stress experiments”) for N6mA profiling. We have revised the text to “we randomly chose 3 pairs of control and stressed mice from three independent stress experiments...”.

We cited three N6mA profiling studies in *Drosophila*, *Xenopus*/mouse kidney and mESCs to indicate that the N6mA-IP-seq utilized in our study has been well established and widely across various genomes including mammals. The specificity of N6mA antibody used in our study has been extensively validated in these publications.

2. *“Pg 8 – statements about GO analyses here are also confusing as well as vague. What does it mean to say “critical neuronal genes were upregulated in response to stress”. What makes a gene critical? And isn't it self evident that the same genes would not be down regulated by stress that were upregulated?”*

We are sorry for the confusion. As described in the Results Section (Page 9), we calculated the overall N6mA reads ratio at significant intragenic gain- or loss-of-N6mA regions of each gene between stressed and control PFC, and further subcategorized them into 4 groups: gain-of-N6mA/upregulated, gain-of-N6mA/downregulated, loss-of-N6mA/upregulated and loss-of-N6mA/downregulated genes. Only upregulated genes with overall intragenic N6mA loss displayed strong association with biological pathways such as neurodevelopment, neuron morphogenesis and transmission of nerve impulse, which have been well documented to contribute to in brain stress response (McEwen et al, 2015; Ressler et al, 2016). These relevant biological terms were not found in other groups of genes (Revised Supplementary Fig. 5a-5c). We filtered the genes based on the RNA-seq expression data (Reads Per Kilobase of transcript per Million mapped reads), and focused on genes with RPKM values more than 0.5 and log₂ fold changes more than 0.1. Genes showed less than the fold change cutoff were considered no change. We have revised the online methods section to make this point clear.

3. *“Pg 10 – what does it mean to say – “Consistent with enhancer regions, N6mA was also depleted at*

H3K27ac, H3K4me1, and P300regions (Supplementary Fig. 5c). However, no significant N6mA changes were observed in these regions” – these two statements seem contradictory.”

It has been documented that the histone modifications, such as H3K27ac and H3K4me1, as well as transcription factor P300 binding sites largely overlap with the cis-regulatory regions/enhancers in a variety of different somatic tissues (Shen et al, 2012; Calo and Wysocka, 2013; Zhu et al, 2013). However, these histone modifications also mark other genomic regions, such as promoters. Bing Ren and colleagues at UCSD took advantage of the enhancer signature chromatin and transcription factor binding pattern, and developed an enhancer predicting algorithms to precisely predict *bona fide* cis-regulatory elements in different tissues and cell types (Shen et al 2012). We first investigated the correlation between N6mA and cortex enhancer regions, predicted by Shen et al. The enrichment of control and stressed N6mA mapped reads were compared with non-enriched input. We found that both control and stressed mice showed significant depletion of N6mA in cortex enhancers relative to baseline input (Revised Supplementary Fig. 6a, t-test, $p < 0.001$). Similar depletion of N6mA was found in cortex-specific enhancers that exclusively identified in cortex, not other somatic tissues (Revised Supplementary Fig. 6a, t-test, $p < 0.001$). To confirm depletion of N6mA in cis-regulatory regions, we directly investigated the N6mA distribution at these enhancer-signature histone marks and p300 binding sites. Depletion of N6mA in enhancers is further strengthened by the similar significant depletion of N6mA in H3K27ac, H3K4me1 and p300-binding sites (Revised Supplementary Fig. 6c and Supplementary Fig. 10a, t-test, $p < 0.001$). Thus, we concluded “Consistent with enhancer regions, N6mA was also depleted at H3K27ac, H3K4me1, and p300 regions”

After finding N6mA was depleted in cis-regulatory regions relative to non-enriched input, we further investigated how restraint stress could dynamically induce N6mA changes in these regions, by directly comparing control and stressed N6mA profiling. As shown in Revised Supplementary Fig. 6b, stress could further significantly (t-test, $p < 0.001$) reduce the N6mA in both cortex enhancers and cortex-specific enhancers, implying altered N6mA upon stress could impact enhancer activities in mouse cortex. Interestingly, although the same trend (further N6mA depletion upon stress) was observed in H3K27Ac, H3K4me1 and P300, the stress-induced N6mA further depletion was not significant (Revised Supplementary Fig. 6d and Supplementary Fig. 10b, $p > 0.1$, t-test). This observation suggested that N6mA dynamic changes in response to stress were more specific to cis-regulatory regions. The statements in our original submission were not contradictory. We have revised the text to make this point clear.

4. *“Figure 1 – there is no need to give precise values within the legend. The dot blots are PCR based but is this the most stringent means for quantifying a difference? Why not qPCR? The results would have been strengthened by including additional brain regions here to confirm effects are specific to prefrontal cortex.”*

The reviewer referred to the quantification of N6mA in Figure 1 with some misunderstanding. We first utilized the stringent and definitive highly sensitive ultra-performance liquid chromatography tandem mass spectrometer (UHPLC-MS/MS) to precisely quantify N6mA dynamic changes upon stress in several brain regions involved in stress response. Interestingly, although detectable N6mA was present in all these brain regions, only N6mA in PFC showed substantial and significant increase from average 6.6 p.p.m. (N6mA per million dA) to 25.5 p.p.m. upon stress (Fig. 1c).

We then employed N6mA dot blots, which directly cross-link equal amount of genomic DNA isolated from PFC of control and stressed mice and detect the overall N6mA abundance using N6mA specific antibody. This approach is not PCR based, but rather provide semi-quantitative validation of HPLC data. In order to validate the specificity of N6mA antibody, we performed PCR reactions with

unmodified or methylated Adenosine in the dNTP mix. The N6mA antibody specifically reacted with PCR products amplified using N6mA-containing dNTP.

5. *“Figure 2 – a) the volcano plot looks strange in that there appears to be a cut-off for the log10p value as well as come restriction that results in a non-continuous distribution. Is this a by-product of the pair-wise replicates?”*

The differential N6mA tests were performed on control and stressed N6mA read counts in 5,451,548 binned mouse genomic regions. Due to the large number of bins and the tests being performed on discrete data (counts), some bins result in similar p-values, which form the “stripes” in the volcano plot. This is the nature of the data, and is not a by-product of the pair-wise replicates. We further validated the N6mA differential regions by another well-established computational algorithm, DESeq2 (see the full response to Reviewer 2 #3).

6. *“Figure 2 – b and c) the heat map of fold change here is also confusing. Is this just replicating what is in the pie charts? How is a fold enrichment of zero the most important?”*

The pie chart was created using significant gain- or loss-of-N6mA regions annotated to the genome. It provided critical information for the dynamic N6mA changes in various genomic regions. The N6mA dynamic changes upon stress predominantly occurred intragenic introns and intergenic regions including various kinds of transposon elements.

The heatmap was generated using fold enrichment calculation versus expected values, which was calculated by the HOMER (Hypergeometric Optimization of Motif EnRichment) suites. It demonstrated the whether the distribution of the annotated regions was significantly more enriched or depleted than expected. The red color represented fold enrichment (>0) over expected values. Based on the heatmap, significant gain-of-N6mA regions were mostly enriched in intergenic regions (such as transposons) over expected values, whereas the significant loss-of-N6mA regions were mostly enriched in intragenic introns.

7. *“Figure 3 creates similar confusion, especially with the pie chart and the heat map using different color coding”*

As discussed above, the pie chart demonstrated the annotated regions to various genomic features, whereas the heatmap showed the enrichment versus expected distributions. The color key in heatmap represented the enrichment fold changes. The color in the pie chart was randomly generated to simply denote different classes of transposon elements. In the effort to make the figure clear to this point, we have rearranged the pie chart legend to clearly indicate their association with the pie chart. In addition, we have put the fold enrichment heatmap into Revised Figure 3c and 3d to clearly separate them from Figure 3a and 3b pie chart.

8. *“Figure 4 – b) It is hard to gloss over the observation that “sulfur compound biosynthetic process” is almost as enriched as those related to “transmission of nerve pulse”. The use of the term “nerve pulse” is also unusual, presumably the authors are referring to the action potential?”*

The GO analyses were performed by PANTHER Enrichment analyses with biological processes (<http://www.geneontology.org/>). The terms were directly imported based on the fold enrichment ranking. The term “nerve pulse” could also be referred to “nerve impulse” and related to stress response. We have revised the labels in the figure.

9. “Lastly, the paper needs judicious editing for English.”

Thanks for the suggestion and we have had the manuscript edited by a technical editor in our institution.

Reviewer #2

1. “The major conclusions were primarily based on genomic sequencing data. Although N6mA-IP-Seq is indeed a reliable approach for detecting genomic N6mA distribution, orthogonal approach is also needed to validate the results. Authors used N6mA-sensitive restriction enzyme DpnI with qPCR, but restriction sites of this enzyme are enriched in *E. coli* genome, but not in mouse. Thus, it is unclear which conclusions can be drawn from this approach. For example, authors can confirm their conclusions with SMRT sequencing, which can detect N6mA directly.”

We thank the reviewer for the suggestion. SMRT-seq usually requires high sequencing coverage to identify modified DNA bases. Genome-wide SMRT-seq has been applied to the *C. elegans* genome (Greer et al, 2015, Cell), and more recently, fungi (Mondo et al, 2017, Nat. Genet) but “it will be difficult to interrogate the large mammalian genome” (Wu et al, 2016, Nature). Wu et al applied SMRT-seq on the H2A.Z deposition regions to detect 6mA using H2A.X ChIP’ed DNA from mouse embryonic stem cells, with the risk to introduce bias. We also communicated with Dr. Gang Fang at Mt. Sinai, who performed the SMRT-seq presented in the Wu et al Nature paper. His thought was that currently it would be challenging to perform SMRT-seq on mammalian genome, which would be the case for our current study.

Restriction-based N6mA digestion has been extensively tested in two recent publications from Dr. Chuan He’s group to identify N6mA loci (Fu et al, Cell, 2015; Luo et al Nature Communication, 2016). Luo et al demonstrated the restriction enzyme DpnI not only recognizes a canonical GATC sequence motif, but also cleaves fully methylated double stranded DNA at CATC/CATG sites (CATC and GATG are complementary to each other), which further expands the application scope of this method. The reviewer is incorrect to state “restriction sites of this enzyme are enriched in *E. coli* genome, but not in mouse”. DpnI has been shown to be an excellent restriction enzyme to digest DNA at fully methylated sites without preference for specific genomes, as long as the adenines are methylated in the context of its recognition sites. This has been demonstrated with green algae genome. In summary, the 6mA-IP-seq and restriction based 6mA testing are currently the only two reliable approaches to validate 6mA in mammalian genome.

We have carefully adopted the published method to validate our sequencing results. We randomly chose 6 significant differential N6mA regions (4 intragenic regions and 2 intergenic regions) from the top 50 differential regions, and DpnI-qPCR results validated the N6mA dynamic changes in these loci upon stress. These data indicate that our sequencing analyses are biologically and statistically reliable.

To further address the reviewer’s comments, we validated ten additional significant gain- or loss-of-N6mA regions upon stress by DpnI digestion and qPCR analyses (Revised Supplementary Fig. 3a, b). Furthermore, we also validated 4 loss-of-N6mA regions on the genes involved in neurodevelopment and behavior from GO analyses (Revised Supplementary Fig. 3c). The qPCR results were faithfully consistent with our sequencing analyses. These new experimental validations, together with new computational analyses with an established algorithm (DESeq2, please see below), strongly support the reproducibility and reliability of our data.

2. *“For the N6mA bioinformatics analysis, authors used “edgeR” based method to get differential deposition regions. Although this is reasonable, the data need to be analyzed by other well-known statistic models which interrogate differential deposition. Otherwise it will be difficult to evaluate the robustness of their conclusions.*

We thank this reviewer’s suggestion. edgeR is arguably the most widely used method for comparing different types of high-throughput sequencing data such as RNA-seq or ChIP-seq (original paper has been cited for 5346 times since 2010). Here we applied edgeR on read counts from 500 bp binned mouse genome to identify differential N6mA regions between control and stressed conditions. The differential N6mA loci were further experimentally validated by DpnI-digestion and qPCR.

Based on the reviewer’s suggestion, we have applied another well-established computational algorithm, DESeq2 (original paper cited 2600 times since 2014), to validate the N6mA differential regions identified by edgeR. By using DESeq2, we identified 50172 gain-of-N6mA regions and 30100 loss-of-N6mA regions upon stress with the similar criteria used in edgeR. We further examined the correlation of the N6mA differential regions identified by these two packages, and obtained consistent results generated by these two packages. For instance, we found 96.9% of loss-of-N6mA regions identified by edgeR in our original submission were overlapped with regions identified by the new DESeq2 analyses ($p < 2.2e-16$, binomial test).

Since we found only upregulated genes with overall intragenic 6mA loss displayed strong association with biological pathways such as neurodevelopment, neuron morphogenesis and transmission of nerve pulse, which have been well documented to play part in brain stress response (McEwen et al, 2015; Ressler et al, 2016). These relevant biological terms were not found in other groups of genes (Revised Supplementary Fig. 5a-5c). Thus, the full overlap of loss-of-N6mA regions identified by different algorithms support our original conclusion on the correlation between N6mA dynamics with gene expressions in response to environmental stress.

In addition, we found 73.4% of gain-of-N6mA regions identified by edgeR were overlapped with regions identified by the DESeq2 analyses ($p < 2.2e-16$, binomial test).

Taken together, we have utilized the two most prominent and well-established computational algorithms to identify the N6mA dynamic regions, and obtained very consistent results.

3. *“The authors mentioned “only 8.8% of loss-of-N6mA regions located on LINEs, while 33.9% enriched on simple repeats” upon stress. In general, quantification at simple repeats are always problematic. Different statistical models may render different results, which reinforces the necessity to analyze the data thoroughly (see above).”*

We have carefully utilized the well-established program and algorithms for the transposon analyses. First, we identified significant N6mA dynamic regions upon stress in mouse PFC from N6mA-IP-seq uniquely mapped reads of replicated samples by edgeR computational algorithms. We then applied the published HOMER suite annotation function (annotatePeaks.pl) to obtain the genomic information of each gain- or loss-of-N6mA regions upon stress. We found substantially stronger enrichment of increased N6mA regions on LINE transposons than loss-of-N6mA regions upon stress. In order to draw a direct correlation between N6mA and transposon expression, we calculated the overall N6mA reads from replicated control and stressed conditions on 103 different classes of LINE transposons bearing significant N6mA dynamic regions. These transposon expressions were obtained from replicated RNA-seq experiments using the recent published Tetranscripts (Jin et al, 2015), and validated by another package piPipes (Han et al, 2015). These two computational packages were the most utilized tools for calculating transposon expression from RNAseq, and rendered the consistent results of LINE expression in our case. We finally correlated the N6mA dynamic changes with LINE expression, and found that the majority of LINE transposons showed concomitantly increased N6mA

and reduced expression. We finally applied the same approaches to analyze SINE transposons, and performed Chi-squared tests to compare these two classes. The increased N6mA and decreased expression were specifically and significantly enriched in LINE but not SINE transposons ($p < 0.001$, Chi-Squared tests). Thus, we have revealed a specific and significant correlation between increased N6mA and reduced LINE expression in mouse PFC upon stress.

4. *“Interestingly, authors found that methylated cytosine (mC) was strongly enriched at loss-of-N6mA regions but only showed modest distributions at gain-of-N6mA regions. This could offer an important mechanistic link of N6mA and 5mC. How is the 5mC level of LINE1s upon stress?”*

This is an excellent suggestion. In order to address this question, we have performed Methylated DNA immunoprecipitation using specific 5mC antibody using replicated control and stressed PFC DNA. Interestingly, MeDIP coupled with qPCR detected a significant increase of methylation in L1td1 upon stress (Revised Supplementary Fig. 8b), suggesting a potential crosstalk between N6mA and 5mC in LINE transposons.

5. *“Why does only PFC show substantial N6mA increase with stress? This seems to be a very interesting observation. The authors could offer some hints or explanation in discussion.”*

The reviewer raised an interesting question. Recent studies from others, as well as our present data, indicated the potential epigenetic roles of N6mA in regulating gene expression, possibly in a tissue or cell type-specific manner. Prefrontal cortex, Amygdala and Hippocampus are three vital regions in mammalian brains in behavior and cognitive function, as well as in regulating the stress response (McEwen et al, 2015). Interestingly, it has been suggested the chronic stress could induce morphological and molecular changes within PFC to render cognitive rigidity and increased vigilance. These changes are PFC specific, and could be conveyed by the rapid alteration of N6mA upon stress. In agree with this, a recent pre-print in bioRxiv reported a specific accumulation of N6mA in the prefrontal cortex during the formation of fear extinction memory. It is important to note that the stress condition applied in the current study only represents one of many unknown environmental paradigms, which could spatially and temporally trigger N6mA dynamics to deal with the environmental changes at the epigenetic level. Thus, further studies will be important to precisely elucidate the roles of N6mA in specific brain regions in response to various stimuli. We have discussed this point in the Discussion section.

6. *“The authors found that 49.2% of intergenic gain-of-N6mA was annotated to long interspersed nuclear element (LINE) transposons, which is the highest enriched transposon over expected. However, the authors need to elaborate on their approach for mapping transposons to the genome: how are they aligned and counted? Using unique mapping reads or assign the reads randomly.”*

Please refer to our response to Point #3.

Reviewer #3:

1. *“What were the parameters used for the identification of the peaks to call them “significant enrichment”? The authors used an independent method by restriction enzyme digestion to validate few of these changes. This should be extended to at least a dozen in total and should comprise genes that show the highest N6mA changes between normal and stress conditions as well as the lowest (to validate their cut-off). I doubt that all the called peaks would be positive in this case.”*

We have discussed a similar point in detail in the response to Reviewer #2-#2. In brief, established computational algorithms (edgeR package in R/Bioconductor environment) were utilized to compare 3 control and 3 stressed normalized N6mA read density in 500bp binned mouse genome (N6mA-seq resolution) considering sample variations. Differential regions with p value < 0.05 (False Discovery Rate < 0.077) were considered significant.

During the revision process, we have applied another well-established computational algorithm, DESeq2 (original paper cited 1914 times since 2014), to validate the N6mA differential regions identified by edgeR. The differential N6mA regions upon stress identified by these two independent programs significantly overlapped, indicating that our data and analyses are highly reproducible. Based on this reviewer's request, we validated ten additional significant gain- or loss-of-N6mA regions upon stress by DpnI digestion and qPCR analyses (Revised Supplementary Fig. 3a,b). Importantly, we also validated several loss-of-N6mA regions on genes involved in neurodevelopment and behavior from GO analyses (Revised Supplementary Fig. 3c). The qPCR results were consistent with our sequencing analyses. These new experimental validations, together with new computational analyses by DESeq2, strongly support the reproducibility and reliability of our data.

2. *““When these” significant peaks” were compared between the treated and non treated samples, what was the fold change cut-off used to claim that regions are enriched or depleted of N6mA. From the fig 2a I can assume that the majority of the regions were enriched or de-enriched with a fold difference of 1.5 or -1.5 (x axis) and only a few regions had a fold enrichment or de-enrichment of >2. Most of the peaks in fig 2d have a scale ranging from 0.6 to 1.2 and it is hard to believe that an enrichment of 1.2 is a real enrichment, especially when the basal level is so low. Please comment on this and justify why the 1.2 enrichment are claimed to be significant. Same problems I see for the de-enrichment in supplementary figs 5 and 6.””*

We agree that the fold changes are not overwhelmingly big for many significant peaks. This is the nature of the data, that is, the dynamics of the signals are not dramatic. These peaks, however, are statistically very significant even after multiple testing correction. We have utilized the two most prominent and well-established computational algorithms to identify the N6mA dynamic regions, and obtained very consistent results. Moreover, we have also applied methylation-sensitive restriction digestion coupled with qPCR to experimentally validate N6mA differential regions upon stress.

Supplementary Fig. 5 (Revised Supplementary Fig. 6) and Supplementary Fig. 6 (Revised Supplementary Fig. 7) were different analyses without the involving cutoffs. These figures essentially aim to demonstrate the dynamic enrichment or depletion of N6mA on various genomic features or epigenetic modification sites. The fold changes of normalized N6mA reads in control and stressed conditions were first calculated against non-enriched input DNA (Revised Supplementary Fig. 6a, c and 7a) in the known genomic regions such as enhancers (Revised Supplementary Fig. 6a) or histone modification sites (Revised Supplementary Fig. 6c). The regions were divided into 100 bins and the statistical differences between control versus input or stress versus input in these 100 bins were determined by Welch's t-test with p-values indicating whether the enrichment or depletion was statistically significant. In addition, the stress-induced dynamic N6mA alterations in these regions were further tested by calculating average fold changes between stressed and control N6mA reads with p-values indicating statistical significance (Revised Supplementary Fig. 6b and 7b).

3. *“In supplementary figs 5 sand 6 the blue color gradient is used very extensively to show the de-enrichment. However this is misleading. When the color-coding is compared to the actual scale bar it is hardly distinguishable from the 1x fold change, suggesting that barely anything is happening. I suggest using either proper scale for a blue color with a maximum corresponding to the red color, or just avoid using it at all.”*

We thank the reviewer for this comment. Please see our discussion above regarding these figures. The imbalanced color key arose from the fold enrichment difference in different genomic regions. The significant enrichment (red color) in Supplementary Fig. 5 (Revised Supplementary Fig. 6) were from the positive control, which indicated the N6mA mapped reads in stressed PFC were specifically and dramatically enriched in gain-of-N6mA regions upon stress identified by edgeR. The enrichment of N6mA on these regions should be the highest among all genomic regions. In comparison, the N6mA reads were specifically depleted on enhancer regions. While the fold changes of N6mA depletion were not as strong as its enrichment on the gain-of-N6mA regions, the depletion in enhancers was indeed statistically significant ($p < 0.001$, t-test).

In our initial submission, the color key was skewed due to that we focused on the regions enriched with N6mA to start with. To address the reviewer's concern, we have separately generated the color key for the enhancer and histone modification regions and positive control regions to clearly demonstrate the fold change differences.

Similarly, we found the specific enrichment of 5mC on the loss-of-N6mA regions upon stress in Supplementary Fig. 6 (Revised Supplementary Fig. 7). We now separately generated the color keys for the gain- or loss-of-N6mA regions to show the fold change differences.

4. *“The authors showed a correlation between m5C and changes in m6A levels upon stress. It would also be informative to have an overall correlation between m5C and m6A in the brain in wild type condition (as well as upon stress if possible).”*

We have found significant correlation between N6mA and 5mC on the dynamic loss-of-N6mA regions upon stress. However, we do not expect the strong correlation between 5mC and N6mA in static levels, especially given the overall low level of N6mA in control mice. We performed genome-wide correlation between N6mA and published 5mC in both non-neuronal and neuronal cells, and found a modest correlation between these conditions (Revised Supplementary Fig. 8a). Interestingly, the global N6mA profile in control PFC demonstrated stronger correlation with 5mC than stressed PFC, and 5mC profile in neuronal cells was closer to N6mA than non-neuronal cells (Revised Supplementary Fig. 8a). These analyses supported a closer correlation between N6mA and 5mC in neuronal cells.

5. *“What is the exact association between N6mA changes upon stress and the genes that are linked to neuropsychiatric disorders shown in fig. 5? Do these genes show loss of m6A and are up-regulated, as shown before for the neuronal genes, or is this association more random, which might suggest some indirect effects.”*

We identified all genes bearing dynamic N6mA changes upon stress, including gain-of-N6mA or loss-of-N6mA, suggesting potential epigenetic roles of N6mA on these genes. We then overlapped the human orthologs of these genes with known GWAS loci associated with neuropsychiatric disorders (loci were obtained from citations in the main text), and found specific and significant correlations (Chi-squared test with p-values indicated). To avoid indirect effects, we generated three random gene lists to match GC content, gene size and neuronal expression, and Pearson's Chi-squared test clearly indicated that these random lists were not correlated with neuropsychiatric disorders. In addition, the GWAS loci involved in Aortic lesion and Obesity were not significantly correlated with N6mA dynamic genes upon stress, which strongly support the link of N6mA to loci related to mental illness.

6. *“In fig 3c the color-coded scale showing the fold change is logarithmic, but the range is relatively small. For better clarity and consistency with other figures it would be better to keep it linear.”*

We utilized log₂ fold changes to illustrate the correlation between N6mA dynamics and LINE expression. The log fold changes are clearly indicated in the color key. The log fold change between -1 to 2 would not be considered small.

7. *“In supplementary fig 5c the authors examined Pol II enrichment (and not P300 as mentioned in the text).”*

We thank the reviewer for pointing out this issue. We indeed investigated the correlation between N6mA and Pol II in this figure, and p300 in Supplementary Fig. 8 (Revised Supplementary Fig. 10). We have revised the text to clarify this.

8. *“P11: figure 5a (not 6a)”*

We have corrected this error.

REVIEWERS' COMMENTS:

Reviewer #1 (Remarks to the Author):

the authors have responded to all my concerns

Reviewer #2 (Remarks to the Author):

In this revised version, the authors have carried out additional experiments and analyses to address the concerns this reviewer raised. In general, these efforts further strengthened the conclusions that 6mA is induced upon stress, thereby repressing L1 transposons. Given the quality of the data and novelty of the results, this reviewer now believes this paper is suitable for publication in principle. However, some important issues still need to be fully addressed in the manuscript.

1. For the answer to question 1, we think the authors misunderstand our points. The DpnI cutting site "GATC" or its derivatives is the N6mA motif in the E. coli genome; but in the mouse genome. This approach potentially biases the results to a subset of DpnI cutting sites that carry N6-mA, instead of the genome-wide N6mA sites. On the other hand, we also agree with the authors that whole genome SMRT sequencing seems to be a challenging undertaking with the current technology. In light of these considerations, we would like to have the author add a few sentences to address the above-mentioned concerns of Dpn1 approach. As a matter of fact, some laboratories have solely relied on the Dpn1 approach to interrogate 6mA deposition and apparently reached wrong conclusions, including another manuscript mentioned by the authors in the rebuttal. Therefore, it is pertinent for the authors to alert the readers about the differences.
2. For question 3, we can't seem to find the response to directly address the issue of simple repeats.

Reviewer #3 (Remarks to the Author):

The authors have addressed all my concerns.

Responses to Reviewers' Comments:

1. *"For the answer to question 1, we think the authors misunderstand our points. The DpnI cutting site "GATC" or its derivatives is the N6mA motif in the E. coli genome; but in the mouse genome. This approach potentially biases the results to a subset of DpnI cutting sites that carry N6-mA, instead of the genome-wide N6mA sites. On the other hand, we also agree with the authors that whole genome SMRT sequencing seems to be a challenging undertaking with the current technology. In light of these considerations, we would like to have the author add a few sentences to address the above-mentioned concerns of Dpn1 approach. As a matter of fact, some laboratories have solely relied on the Dpn1 approach to interrogate 6mA deposition and apparently reached wrong conclusions, including another manuscript mentioned by the authors in the rebuttal. Therefore, it is pertinent for the authors to alert the readers about the differences."*

We thank the reviewer for the clarification. We agree that the DpnI can cleave fully methylated adenine at GATC/CATC/CATG, which are not the only sites containing methylated adenines in mouse genome. However, this method indeed provides a great alternative to independently validate differential methylated adenines within the DpnI cutting sites detected by 6mA-IP, supporting the reproducibility and reliability of our 6mA-IP-seq. Future methodology development is required to comprehensively and unbiasedly map 6mA at single-base resolution. We have added this comment in the revised Result section.

2. *"For question 3, we can't seem to find the response to directly address the issue of simple repeats."*

We apologize for the confusion. We do agree with the reviewer that "In general, quantification at simple repeats are always problematic." However, the "only 8.8% of loss-of-N6mA regions located on LINEs, while 33.9% enriched on simple repeats" defined in our study was not quantification of simple repeats. We used published HOMER software suite to annotate the differential 6mA regions to various genomic features. We found 8.8% of loss-of-6mA peaks upon stress mapped to the LINE elements, and 33.9% of these 6mA dynamic peaks annotated to the loci with simple repeats, which is not the focus of this paper.